# Basic Guide for Approaching Drug Delivery with Extracellular Vesicles

**DOI:** 10.3390/ijms251910401

**Published:** 2024-09-27

**Authors:** Sergey Brezgin, Oleg Danilik, Alexandra Yudaeva, Artyom Kachanov, Anastasiya Kostyusheva, Ivan Karandashov, Natalia Ponomareva, Andrey A. Zamyatnin, Alessandro Parodi, Vladimir Chulanov, Dmitry Kostyushev

**Affiliations:** 1Laboratory of Genetic Technologies, Martsinovsky Institute of Medical Parasitology, Tropical and Vector-Borne Diseases, First Moscow State Medical University (Sechenov University), 119991 Moscow, Russia; seegez@mail.ru (S.B.); aleksa.yudaeva@gmail.com (A.Y.); kachanov.av99@gmail.com (A.K.); kostyusheva_ap@mail.ru (A.K.); ivan.karandashov@gmail.com (I.K.); ponomareva.n.i13@yandex.ru (N.P.); 2Division of Biotechnology, Sirius University of Science and Technology, 354340 Sochi, Russia; parodi.a@talantiuspeh.ru; 3Department of Pharmaceutical and Toxicological Chemistry, First Moscow State Medical University (Sechenov University), 119146 Moscow, Russia; oleg.danilik7@gmail.com; 4Faculty of Bioengineering and Bioinformatics, Lomonosov Moscow State University, 119234 Moscow, Russia; zamyat@belozersky.msu.ru; 5Belozersky Institute of Physico-Chemical Biology, Lomonosov Moscow State University, 119992 Moscow, Russia; 6Department of Biological Chemistry, Sechenov First Moscow State Medical University, Trubetskaya Str. 8-2, 119991 Moscow, Russia; 7Department of Infectious Diseases, First Moscow State Medical University (Sechenov University), 119991 Moscow, Russia; vladimir@chulanov.ru

**Keywords:** targeted drug delivery, nanotherapeutics, surface display

## Abstract

Extracellular vesicles (EVs) are natural carriers of biomolecules that play a crucial role in cell-to-cell communication and tissue homeostasis under normal and pathological conditions, including inflammatory diseases and cancer. Since the discovery of the pro-regenerative and immune-modulating properties of EVs, EV-based therapeutics have entered clinical trials for conditions such as myocardial infarction and autoimmune diseases, among others. Due to their unique advantages—such as superior bioavailability, substantial packaging capacity, and the ability to traverse biological barriers—EVs are regarded as a promising platform for targeted drug delivery. However, achieving a sufficient accumulation of therapeutic agents at the target site necessitates a larger quantity of EVs per dose compared to using EVs as standalone drugs. This challenge can be addressed by administering larger doses of EVs, increasing the drug dosage per administration, or enhancing the selective accumulation of EVs at target cells. In this review, we will discuss methods to improve the isolation and purification of EVs, approaches to enhance cargo packaging—including proteins, RNAs, and small-molecule drugs—and technologies for displaying targeting ligands on the surface of EVs to facilitate improved targeting. Ultimately, this guide can be applied to the development of novel classes of EV-based therapeutics and to overcoming existing technological challenges.

## 1. Introduction

Extracellular vesicles (EVs), according to MISEV2023, are “Particles that are released from cells, delimited by a lipid bilayer, that cannot replicate on their own” [1,2]. EVs have emerged as a superior delivery system, offering biological compatibility, biodegradability, low toxicity, and immunogenicity [3]. Furthermore, EVs can cross many biological barriers (e.g., blood–brain barrier [4]) and deliver nucleic acids, proteins, small-molecule compounds, and other cargo [5].

Extracellular vesicles (EVs) are considered a more “natural” delivery system, as they are derived from cells and possess specific surface markers such as CD47, which functions as a “don’t eat me” signal that inhibits phagocytosis by macrophages [6,7]. This signaling mechanism leads to the enhanced retention of EVs in circulation compared to synthetic nanocarriers like liposomes. Furthermore, EVs produced from mesenchymal stem cells exhibit additional properties, including improved targeting to inflamed tissues and immunoregulatory capabilities [8]. Several EV-based therapeutics have entered phases III-IV of clinical trials, and they may receive approval for medical applications in the near future [9]. However, the manufacturing of EVs in accordance with good manufacturing practice (GMP) principles is hindered by factors such as batch-to-batch variability and a lack of standardization in EV research [10,11]. Additionally, the isolation and purification processes for EVs are costly, complex, and yield low quantities, making it challenging to scale up EV production. The application of EVs as drug delivery vehicles also necessitates effective strategies for encapsulating various cargo types (such as small-molecule drugs, RNA, proteins, and others) and ensuring targeted delivery to specific tissues through the incorporation of targeting molecules. Some of these issues have been addressed over the past decade, resulting in effective protocols for the cost-efficient production of EVs and the incorporation of desired cargo for delivery, including industrial-scale manufacturing [12]. Several previous reviews extensively described the different aspects of EV biogenesis, production, and purification for fabricating formulations with immune-regulatory and pro-regenerative properties [12,13,14]. In this review, we focus on providing an up-to-date, optimized, technical workflow for developing EV-based drug formulations, and we summarize methods for the production and purification of EVs, as well as techniques for cargo loading. Additionally, we will discuss the primary approaches for the presentation of targeting ligands in EV-based drug delivery systems and describe the current status of clinical trials of EV products containing loaded therapeutic cargo. Overall, this manuscript aims to serve as a basic guide for developing novel EV-based therapeutics and solving the key fundamental and technological challenges at different stages of drug development.

## 2. EV Biogenesis

EVs represent a group of heterogeneous vesicles that are generally divided into three subcategories based on their biogenesis: exosomes (30–120 nm), ectosomes (30 nm–1 µm), and apoptotic bodies (100 nm–5 µm) [15]. At the same type, the classification of EVs is evolving, and many distinct types of EVs were characterized. The current landscape of EVs includes small high-density lipoprotein (HLD), low density lipoprotein (LDL), supermeres, intermediate-density lipoprotein (IDL), exomeres, VLDL and vaults with a size of 5–72 nm, exosomes, arrestin domain-containing protein 1-mediated microvesicle (ARMM), small ectosomes, and supramolecular attack particles (SMAPs), which range in size from 30 to 150 nm. Larger EVs with a size from 70 nm to several µm include chylomicron remnants, chylomicrons, microvesicles, apoptotic bodies or vesicles, migrasomes, large oncosomes, and exophers. These types have distinct sizes, biochemical composition, and morphology [15]. Exosomes are formed by the inward budding of multivesicular bodies that later fuse with the cell surface, resulting in vesicle release. Ectosomes bud directly from the plasma membrane, and apoptotic EVs form during apoptosis by cell fragmentation. Recently, more subtypes have been discovered, such as migrasomes [16] and exophers [17]; their capacity to specifically deliver molecules for therapeutic purposes is yet to be discovered (Figure 1). The EVs mentioned below mostly represent particles < 1000 nm in diameter, like exosomes and the classic type of ectosomes, also known as microvesicles.

EVs are naturally released from recipient cells into the extracellular space. However, after production, EVs are contaminated with cells, host cell DNA, host proteins, and cell culture media supplements (e.g., fetal bovine serum [FBS]), and thus must be isolated and purified from cell culture media for further use. As EVs are usually manufactured using adherent cells (e.g., stem cells), they can be easily isolated and purified in one step by processing the conditioned media [18].

## 3. Purification Methods

### 3.1. Size-Based Approaches

Plenty of protocols exist for EV isolation, including several combined methods which are summarized and compared in Table 1 [19,20]. The gold standard for EV separation is differential ultracentrifugation (dUC or UC), which combines serial centrifugation steps (Figure 2A). A standard protocol comprises centrifugation at low speed (300× *g*) to remove cells; at medium speed (2000× *g*) to remove larger cell debris; centrifugation at high speed (10,000× *g*) or filtration through a 0.22 μm filter to remove large non-exosomal EVs; and two long, high-speed centrifugation (100,000× *g*; 60–70 min) steps to pellet and wash EVs [21]. dUC allows a high concentration of EVs, but its application for industrial-scale EV manufacturing is complicated [22]. Currently, dUC is the best-characterized method that is used in roughly half the studies related to EVs [23]. However, the separation of EVs with dUC has many disadvantages, including vesicle aggregation [24], disruption [25] and loss [26], contamination with free proteins and their aggregates [27], and a low isolation yield (~25–30%) [28]. Moreover, dUC is time-consuming and has high equipment requirements. EV pelleting efficiency is dependent on parameters such as sample viscosity and centrifuge acceleration, rotor type, and characteristics [22]. Intra- and interlaboratory variability, as well as poor scalability, make following GMP principles difficult with this method [29].

Density gradient ultracentrifugation (DGU) is a variation of dUC that uses a medium (usually a solution of sucrose or iodixanol [30]) of gradated density (Figure 2B). During the centrifugation process, EVs migrate to their equilibrium fractions according to their size, shape, and density. DGU offers the highest EV purity with almost no protein contamination [31], as well as reduces vesicle damage and aggregation [12], but the method is tedious, expensive, low yield, and requires post-purification from gradient media, thus often resulting in considerable EV loss [22].

During ultrafiltration (UF), a sample is passed through a semi-permeable membrane, so small contaminants (below the membrane pore size cut-off) are removed into the filtrate, whereas the desired particles are collected in the retentate (Figure 3A). To remove proteins and small molecules, cut-offs of ~300–750 kDa (~30–75 nm) are a common choice, whereas filters with a higher molecular weight cut-off (i.e., 1000 kDa) can be useful to remove some smaller EVs [2]. UF is preferred by many researchers as it is fast, simple, and does not require expensive equipment [32]. UF is considered time-saving and cost-effective; it allows for concentrating EVs, offering a greater isolation yield than dUC [33]. Also, the UF method is established in the industry for protein production and is therefore easily scalable [34]. However, larger particles that cannot migrate through the membrane can accumulate on the filter upstream surface and block the filtration of smaller molecules, resulting in a “cake” formation effect that decreases sample purity [12,35]. Other disadvantages of UF are possible EV deformation [22], non-specific adsorption of EVs on the filter surface, and filter pore clogging [36].

To reduce the “cake” formation effect, asymmetric depth filtration (ADF) has been proposed for EV isolation [37]. Depth filters use a porous filtration medium with larger diameter pores to retain particles throughout the medium, rather than on the filter’s surface. The filter pores have a tortuous geometry, which allows them to capture larger particles (Figure 3B). Thanks to this setting, soluble components and small molecules are eluted first, while larger components elute later or get trapped within the filter. DF is a simple, fast, and inexpensive method that provides less filter clogging since the entire filter medium participates in fractionation, rather than just its surface.

Tangential flow filtration (TFF), also known as cross-flow filtration, is a more sophisticated version of UF that is based on the same principle. However, in the TFF system, the stream flows tangentially across the UF membrane and a second flow is directed through the membrane (Figure 3C). Smaller particles are filtered through the membrane, and larger particles are flushed from the filter by tangential flow. TFF is a cyclic process, and filtration can be repeated several times to increase sample purity or perform buffer exchange. TFF is time-saving, reduces material deposition, provides high EV yields, and allows for vesicle concentration [38,39,40]. TFF can be used to isolate EVs from large volumes (thousands of liters), and it is established as a scalable industrial approach [39]. However, TFF can result in vesicle contamination with large proteins and DNA [18,41], and therefore often requires a second process (e.g., size exclusion chromatography) to obtain optimal EV purity [40].

Size exclusion chromatography (SEC) is another prominent method that separates particles by size as they pass through a column filled with porous beads [42]. Components larger than the pore size are unable to enter the pores of the column and thus elute first, while smaller molecules or particles become trapped inside the beads and elute later. This method efficiently eliminates smaller molecules, proteins, and small particles in combination with mild processing conditions that preserve the vesicle structure [43]. SEC provides a high purity of isolated EVs, but it is a laborious process that may require specialized equipment [44,45]. Although SEC may be used for scaled preparations, it requires a pre-concentration step for the processing of large amounts of EV-enriched media [46] and has a relatively low yield [47].

### 3.2. Precipitation-Based Methods

Precipitation-based methods (Figure 4A) commonly utilize hydrophilic (e.g., polyethylene glycol [PEG]) or cationic polymers (e.g., protamine) to induce reversible EV aggregation, allowing their separation by centrifugal forces or filtration [46]. Many commercial isolation kits are based on precipitation (ExoQuick, Exo-Prep, Total Exosome Isolation Kit) due to its simplicity and low EV damage [48]. Precipitation-based methods do not require costly or specialized equipment, and they can be applied to large sample volumes required to obtain preparative EV amounts [49]. However, precipitation-based methods are costly and result in high amounts of contaminants, including proteins, that co-precipitate with the vesicles, reducing sample purity. Polymer residues must also be removed, which can lead to EV loss [18]. These disadvantages make precipitation-based methods unsuitable for use in clinical therapy without additional purification steps [50].

### 3.3. Affinity-Based Approaches

EVs can be isolated via immunoaffinity methods based on EV-specific membrane protein molecules, such as CD9, CD63, and CD81 [51]. In immunoaffinity-based isolation, antibodies are usually attached to magnetic beads to form antibody-conjugated magnetic beads that can pull EV populations from crude material (Figure 4B) [18]. Immunoaffinity methods offer high purity [52] and reduce contamination with soluble proteins [35]. However, EV markers are not present on every EV subtype [53]; for instance, only a small number of EVs are positive for both CD63 and CD81 [54]. Moreover, the interaction between the antibody and the EV marker is not easily disrupted. Harsh elution conditions may destroy the EV structure [55].

Recently, a protocol named EV-Elute has been proposed, which allows the recovery of over 70% of bound EVs from antibody-conjugated magnetic beads without compromising EV integrity [56]. This is achieved by optimizing components of the elution buffer and conditions used for the elution process. Other disadvantages of immunoaffinity methods include high antibody costs, relatively low EV yields [35], and method difficulties for isolation from large quantities [57]. On the other hand, aptamers can be used instead of antibodies to lower the cost [58]; aptamer–marker interactions are also easily reversible in mild conditions that allow particle isolation that preserves their integrity.

### 3.4. Chromatography Approaches

To obtain high-purity EVs, multimodal flowthrough chromatography (MFC; also known as bind–elute chromatography) has been introduced [59]. MFC combines size-exclusion and bind–elute chromatography. MFC resins have an inert shell permeated with size-selective pores that surround an absorptive core. When small molecules and proteins enter the pores, they bind to the absorptive core via hydrophobic and charge interactions. Other molecules larger than the size of the selective pores, including EVs, are excluded from the shell and can be collected in the flowthrough. MFC allows us to isolate EVs with a higher purity (comparable to 2D culture and SEC), even from highly contaminated bioreactor preparations, and with a negligible sample dilution compared to SEC [59].

Anion-exchange chromatography (AIEX) is another chromatography-based method that is applied for EV isolation. EVs can be separated by binding to the positively charged column due to negatively charged phospholipids found in EV membranes [60]. AIEX can be scaled to an industrial level and shows a higher sample yield, purity, and size distribution than dUC [61]. Despite its advantages, AIEX can provide only limited specificity due to co-isolation of different EV subtypes and sample contamination with other negatively charged molecules (nucleic acids, proteins, and others) [62,63], in addition to potential damage to the vesicles due to harsh conditions (e.g., acidic buffers) [43].

A combination of methods is also used to isolate highly purified EVs; indeed, ~60% of current studies combine several methods for EV isolation [64]. Combined EV isolation assures better purity and specificity, rather than single isolation methods [34,65], and the shortcomings of one approach can be addressed by another method. At the moment, the most optimal methods are SEC in combination with either TFF or UF. As aforementioned, both TFF and UF are able to concentrate the samples but yield less pure EV fractions, whereas SEC can result in a relatively high EV purity but dilutes the sample. By combining SEC with TFF or UF, highly concentrated EV fractions with an even higher purity can be obtained [13]. Despite the advantages, every additional purification step increases the processing time and reduces the overall EV yield [66].

The methods described have been well-established through years of EV research, and many of them demonstrate a significant potential for scalability, including UF, TFF, and chromatography techniques. In recent years, novel methods for EV isolation for diagnostic purposes have been developed, including microfluidic approaches [67,68,69], asymmetrical flow field-flow fractionation [70], electrophoretic isolation [71], acoustic approaches [72], and others. Despite the high recovery rate, novel methods are not suitable for large-scale manufacturing, and therefore are not considered in this review. Novel methods are described in detail in [35,73].

**Table 1 ijms-25-10401-t001:** Comparison of different EV isolation approaches.

Method	Principle	Scalability	Yield (Recovery)	EV Damage	Purity	Equipment Requirement	Cost	Additional Pre/Post-Steps	Time	Ref.
Differential ultracentrifugation (dUC)	Serial UC steps	+	+ (5–30%)	↑↑↑	++	+++	+	No	↑↑↑	[22,24,25,26,27,28,29]
Density gradient ultracentrifugation (DGU)	Separation of EVs by density using gradient medium	+	+ (5–30%)	↑	++++	+++	++	Yes (media removal)	↑↑↑	[12,22,31]
Ultrafiltration (UF)	Filtration through semi-permeable membranes	+++	+++ (30–80%)	↑↑	+++	++	++	No	↑/↑↑	[12,22,32,33,34,35,36]
Asymmetric depth filtration (DF)	Filtration through porous medium	++	++ (40–60%)	↑	+++	++	++	No	↑↑	[37]
Tangential flow filtration (TFF)	Cross-flow filtration through membranes	++++	+++ (up to 90%)	↑	+++	++	++	No	↑↑	[12,18,38,39,40,41]
Precipitation approaches	EV sedimentation using polymers	+++	++++ (up to 90–95%)	↑↑/↑↑↑	+	+	++	Yes (polymer removal)	↑↑↑	[12,18,48,49,50]
Affinity-based isolation	EV capture via specific interactions with EV markers	+/++	++ (50–70%)	↑/↑↑	++++	++	+++	No	↑↑	[35,52,53,54,55,57]
Size exclusion chromatography (SEC)	Separation by size through a bead-filled column	+++ (combined with UF/TFF)	++ (40–75%)	↑	+++	++	+	Yes (pre-Concentration)	↑↑	[43,44,45,46,47]
Multimodal flowthrough chromatography (MFC)	Combination of size-exclusion and bind–elute chromatography	+++ (combined with pre-concentration)	++/+++ (up to 80%)	↑	++++	++	++	Yes (pre-Concentration)	↑↑	[59]
Anion-exchange chromatography (AIEX)	Binding of EVs to positively charged column	+++	+++ (40–90%)	↑↑	++	++	++	Yes (buffer exchange)	↑↑↑	[43,61,62,63]

The symbols “+” or “↑” represent the lowest quality, while “++++” or “↑↑↑↑” indicate the highest quality.

### 3.5. Challenges of EV Preparation

A low EV yield is a major problem for EV research. Poor EV yields occur for two reasons: (1) small amounts of EVs produced by the cells and (2) EV loss during isolation. One liter of conditioned culture media yields approximately 10^9^–10^11^ EVs using dUC, which is usually enough for only one experiment in one mouse in preclinical studies [74]. The expected number of EVs needed for therapeutic applications is ~10^13^ EVs per dose per patient [46]. There are many methods to increase EV production by cells, such as mechanical stimulation, serum deprivation, hypoxia, and supplementation with small-molecule drugs and additives [12], but these approaches can be associated with cell damage and reduced EV quality. Alternative approaches to enhancing the extracellular vesicle (EV) yield include electrical stimulation and biological transfection. These methods enable the scaling up of EV production while minimizing severe damage to producer cells by stimulating the expression of genes that influence EV production. Cellular nanoelectroporation [75] and biological transfection [76] can also provide high endogenous loading of cargo such as functional mRNA. At the same time, using dynamic and static three-dimensional cell cultures can substantially boost EV production in a scalable manner [77]. The culturing of cells as a spheroids [78,79], the use of hollow-fiber bioreactors [80,81], semi-permeable capsules [82], culturing on microcarriers in stirred tank bioreactors [83] and rational media supplementation (increased glucose, additional nutrients, and others [84]) boosts the EV yield up to 100-fold compared to standard two-dimensional culturing. However, low yields after isolation are still an issue.

These issues have led researchers to explore alternative EV sources, such as EVs derived from bovine milk or plants. While these sources allow a robust production of EVs with high yields, they cannot substitute human EVs for all applications and can cause adverse and unpredictable effects [13]. Non-human EVs can be immunogenic or allergenic depending on the administration route, dosage, and number/frequency of doses [85]. The safety of these materials must be validated for each individual case.

There are currently no methods for simultaneously achieving a high EV yield and high purity. Highly purifying techniques always lower EV yields [86]. Sufficient separation from co-purifying components is a challenge because EVs are contaminated with host nucleic acids, and proteins and their aggregates. Carbohydrates (e.g., hyaluronic acid) can be overlooked contaminants as well [87]. Another possible contaminant is FBS, a common supplement for cell culturing, which is high in endogenous EVs [88]. However, serum-free conditions can cause stress-induced phenotypic changes in cultured cells, promoting the release of EVs containing reactive oxygen species (ROS) and stress-related proteins [89]. Human platelet lysate is often used as a substitute for FBS, as it is a safe and effective supplement that creates a xeno-free environment suitable for culturing various cell types, while reducing the immunologic risks associated with FBS [90]. On the other hand, the variability of platelet concentrations and the presence of additional bioactive factors in human platelet lysate may affect cell behavior and EV characteristics, compromising the reproducibility and standardization of prepared EVs. Nevertheless, some contaminants can supply EVs with different traits; for example, hyaluronic acid has immunomodulatory properties that may be associated with successful EV applications in cancer therapy and immunotherapy [91]. Viral contamination must also be monitored, especially for clinical treatment, because viral particles and EVs are of a similar size [18].

Separating specific EV subpopulations from cell culture for further analysis is still a problem. The heterogeneity of EVs poses a challenge to their application as therapeutic carriers in clinical treatments. For instance, in one study, three EV subpopulations with distinct sizes, protein marker contents, morphologies, and different functional properties were separated using SEC, demonstrating similar results across different cell types [92].

Additionally, appropriate conditions for storing, handling, and transporting EVs must be determined. One comprehensive study showed that freezing EVs in phosphate-buffered saline (PBS) greatly reduced EV recovery, and proposed PBS supplemented with human serum albumin and trehalose as the optimal EV storage method [93]. Other recent studies have utilized EV lyophilization to facilitate handling and transportation, and to prolong the shelf life of the final product [94,95]. However, further exploration of effects of lyophilization on the integrity of EV membranes is necessary [96]. Finally, different storage conditions may be optimal for EVs from different cell sources or for EV subpopulations [97].

The scalable manufacturing and batch-to-batch variability of EVs are other crucial issues facing clinical therapy. To address these, isolation methods suitable for large-scale applications, such as TFF, SEC, or MFC, can be applied. Bioreactors can be used to produce therapeutically relevant concentrations of EVs while maintaining consistent particle sizes and phenotypes [98,99]. As an optimal option, conditioned media after the three-dimensional culturing of cells in a bioreactor can be processed by tandem isolation approaches such as UF-SEC [100] or TFF-SEC [101] to combine high and scalable yields of UF or TFF, with the effective elimination of the main contaminants by SEC.

Another problem is the biomolecular (protein) corona and the effects of different isolation methods on its composition. The surface of most nanoparticles (NPs) is well known to play a pivotal role in their behavior and functionality. The biomolecular corona can be characterized with a spontaneous adsorption of proteins, lipids, sugar moieties, nucleic acids, and metabolites onto NP surfaces [102] due to their interaction with NP surface molecules in biological fluids. Notably, the corona is formed rapidly (<30 s) [103]. After corona formation, NPs’ physicochemical properties, targeting abilities, and biological responses are dictated by the adsorbed molecules [102,104], posing difficulties for predicting how the biomolecular corona may affect NP activity in vivo due to its dynamic nature. Coronas form on the surface of different kinds of NPs, including lipid-based NPs [105,106], polymeric NPs [107], and inorganic NPs [108]. Recently, biomolecular coronas were shown to be generated around EVs as well [109]. Additionally, it was demonstrated that the isolation method can deform the biomolecular corona: UC and TFF may change its composition and abrogate EV function [110].

## 4. EV Cargo Loading Methods

The application of EVs as a delivery platform requires efficient methods for packaging therapeutic cargo into the vesicles. Payload loading can be endogenous (based on the natural ability of cells to form EVs) or exogenous (loading molecules into EVs after the EVs have been isolated). Whether endogenous or exogenous loading methods are used, the cargo can be loaded either passively or actively.

During passive loading, cargo is simply incubated with donor cells or directly with EVs. This technique can be used to load small molecules, particularly hydrophobic small-molecule drugs. However, the incubation method is rarely used due to its low loading efficiency [111] and poor reproducibility [112]. Other methods are categorized as active loading techniques, and include sonication, electroporation, freeze-thawing, transfection, and extrusion.

### 4.1. Endogenous Loading

Endogenous loading, or pre-loading, is an approach based on the ability of donor cells to incorporate molecules into EVs during EV biogenesis (Figure 5). Several methods of endogenous loading are used for RNA, protein and small molecules packaging, including simple incubation, the application of EV-associated motifs, the interaction of target molecules with the plasma membrane of membrane proteins, viral protein-assisted loading, and others. The main advantage of endogenous loading is the preservation of EV membrane integrity [113].

#### 4.1.1. Cells Transfection/Transduction for RNA Loading

This approach often utilizes transfecting or transducing producer cells with vectors encoding therapeutic molecules (or with therapeutic molecules directly, for example, with siRNA), but cells can also be simply co-incubated with putative cargo for this purpose. One drawback of using transfection for EV loading is the potential alteration of the structure and properties of EVs by the transfection reagents [114]. Another concern is cytotoxicity and difficulty to completely remove the transfection reagent [115,116]. Furthermore, transfection reagents can form lipid micelles with loaded molecules that are similar in size to, and imitate, cargo-loaded EVs, leading to possible sample contamination [117]. Additionally, transfection reagents can affect EV cargo delivery to recipient cells [118,119]. Physical transfection, called electrical stimulation, can also be used for the endogenous loading of diverse cargoes. For example, Y. Ma et al. used nanoelectroporation to load human bone morphogenetic protein 2 (BMP-2) and human vascular endothelial growth factor A (VEGF-A) mRNAs into EVs. The nanoelectroporation principle is based on the seeding of cells onto a track-etched membrane with nanopores (400 nm), and the loading of PBS supplemented with desired plasmids under the membrane. The application of electric fields across the membrane induce the transfer of plasmid into cells via nanopores, resulting in efficient transfection and relatively low toxicity [75]. A similar principle was used to load EVs secreted by dermal fibroblasts with COL1A1 RNA for the treatment of photoaged skin [120]. Nevertheless, the scalability of the method requires additional investigation. Although endogenous loading appears to be less effective in terms of loading small molecules, it seems to be more suitable for loading larger molecules [121].

#### 4.1.2. EV-Associated Motifs for RNA Loading

EV-associated motifs (EXOmotifs or “zipcodes”) are special motifs usually found in 3′-untranslated regions (3′-UTRs) of RNA. They interact with EV biogenesis machinery and help with RNA loading into EVs. EXOmotifs either recruit RNA-binding proteins (RBPs) that mediate the packaging of therapeutic RNA into EVs, or possibly modulate direct RNA interaction with lipid membranes during EV biogenesis [122]. EV-associated motifs have been found in both mRNAs [123] and microRNAs (miRNAs) [124]. hnRNPA2B1 ribonucleoprotein (RNP) can directly interact with GGAG motifs and control miRNA sorting [125], whereas the SYNCRIP protein is able to bind to GGCU motifs of miRNA [126]. EXOmotifs can direct RNA of interest to be packaged inside EVs. For instance, miRNA targeting the ATXN3 gene was engineered using ExoMotif GGAG, resulting in its 3-fold enrichment in EVs [127]. However, the loading process tends to be tissue-specific and may be influenced by secondary and tertiary RNA structures [122,127].

The Psi (Ψ+) signal is an alternative to EV-associated motifs. It can be found in retrovirus genomes (e.g., of human immunodeficiency virus [HIV]) and acts as a packaging signal for the retroviral RNA genome [128]. This signal specifically interacts with the Gag protein and allows the transport of RNA into virus-like EV-mimetic particles (VLPs) that are structurally similar to EVs. Inserting a Ψ+ signal into the RNA sequence can allow RNA loading into VLPs. One study used the nanomembrane-derived extracellular vesicles for the delivery of the macromolecular cargo (NanoMEDIC) system for single guide RNA (sgRNA) packaging by incorporating Ψ+, along with the Tat activation response element (TAR) to increase the expression of CRISPR-Cas sgRNA [129]. To liberate sgRNA from the long mRNA transcript, self-cleaving ribozymes from hammerhead and the hepatitis delta virus flanked the sgRNA sequence. The Ψ+-mediated packaging loaded sgRNA three times more efficiently than stochastic loading. The main drawback of this system is the use of Gag and Tat proteins, which are associated with oxidative stress and can induce cellular toxicity and immunogenicity when used in vivo [130].

#### 4.1.3. Interaction of RNA with Proteins Enriched within EV Membranes

Another strategy for packaging RNA into EVs is through indirect interactions of the desired molecules with EV membrane proteins. For this purpose, RNA-binding proteins (RBPs) can be fused to proteins that are abundant in EVs, like tetraspanins CD9, CD63, and CD81. Then, RBPs can specifically bind to the sequence and/or structural motifs of RNA via one or several RNA-binding domains (RBDs), thereby packaging RNA into EVs [131]. For instance, CD9 was fused with human antigen R (HuR) RBP that mediates the interaction between AU-rich elements (AREs) in the target RNA and HuR [132]. The loaded RNA was localized inside EVs, increasing the loading of Cas9 mRNA and miR-155 by up to 10-fold and 7-fold, respectively.

Several studies have utilized CD63-mediated packaging. The targeted and modular EV loading (TAMEL) approach was proposed as a platform for RNA loading [133]. Here, RNA is engineered with MS2 stem loops that can specifically interact with MS2 bacteriophage coat protein (MCP) fused to CD63 protein. This technique improves RNA packaging into EVs up to 6-fold, but a low endosomal escape of loaded RNAs was observed. An alternative approach uses a specific Com/com interaction between the inserted RNA aptamer com and Com protein fused to CD63 [134]. By additionally overexpressing vesicular stomatitis virus protein G (VSV-G) in the donor cells, EVs were loaded with CRISPR-Cas RNP complexes and demonstrated efficient endosomal escape and gene editing, up to 10-fold higher than EVs without an RNP-enriching mechanism. Viral VSV-G protein increases EVs’ secretion and provides a broad tissue tropism of particles, as many types of cells express the receptor for VSV-G. The main disadvantage of packaging using 3′-UTR-mediated mRNA modifications is the activation of nonsense-mediated RNA decay (NMD) that causes destabilization of mRNA and its degradation in both packaging cells and acceptor cells [135,136].

To overcome the obstacles of low mRNA stability and insufficient loading, a novel TAMEL-based platform has been developed [137]. The authors used an engineered RBD derived from Pumilio and the FBF homology domain (PUFe) fused to CD63, while the target RNA was engineered with 3′-UTR repeats to specifically interact with the RBD. CD63-PUFe showed 1.7-fold to 4.3-fold better loading efficiency than CD63-MCP in three independent experiments, suggesting that the RBD choice and design influence the efficiency of RNA loading. Additionally, the co-expressed VSV-G and cytoplasmic poly(A)-binding protein (PABPc) improves EV cargo delivery and ensures a better stability of mRNA, respectively. PABPc binds to the poly(A)-tail of mRNA and protects it from degradation through the NMD pathway. Indeed, the addition of PABPc improved mRNA enrichment in EVs: the PABPc/mock ratio varied from 3- to 14-fold [137]. Nevertheless, the absolute number of loaded RNA per EV may still be insufficient to provide clinically relevant effects.

Another RNA-RBP interaction was used as the basis for a two-component system called exosomal transfer into cells (EXOtic). CD63 was fused with the L7Ae archaeal ribosomal protein that can specifically interact with a cognate C/D box of the engineered cargo RNA [76]. Connexin 43 protein, a cytosolic delivery helper enriched in EVs, or the more effective mutant S368A protein, was utilized to facilitate the fusion of EVs with the endosomal membrane or the formation of channels that allow the escape of EV content from the endosomal compartment. By using the EXOtic system, catalase mRNA was successfully delivered into cells, reducing the production of ROS and cell death induced by neurotoxin in vivo. Additionally, the transfection of EV producing cells with “booster” plasmid results in an evident increase of EV secretion (up to 15–40-fold).

Moreover, another approach can be applied for the packaging of sgRNA, which is a component of a genome editing system called CRISPR-Cas. sgRNA can be packaged passively due to the formation of sgRNA-Cas9 complexes. For instance, the coupled fusion of GFP with CD63 and the fusion of GFP-specific nanobodies with Cas9 leads to the formation of RNP complexes and their subsequent loading into EVs, but with unsatisfactory efficiency [138]. Notably, Cas9 mRNA is undetectable in EVs when mRNA is used instead of sgRNA, suggesting that sgRNA packaging into EVs is a result of its co-recruitment as a passenger in a complex with Cas9 [139].

#### 4.1.4. RNA Enrichment on the Plasma Membrane

An alternative method for loading endogenous RNA uses a light-inducible dimerization system based on CRY2-CIB1 and MS2–MCP interaction modules [140]. The CIBN (a truncated version of CIB1) protein is fused to a palmitoylation sequence (palm) that induces protein enrichment in the cell membrane, while CRY2 is conjugated to MCP protein. The application of blue light induces the reversible binding of membrane-enriched CIBN with CRY2-MCP, and the subsequent accumulation of MS2-RNA on the plasma membrane via MCP-MS2 interaction. Thus, the amount of loaded miRNA is 14-fold higher than in EVs obtained from cells cultured without light illumination. However, the system relies on RNA loading during EV biogenesis [140], and blue light illumination may affect cell viability through ROS generation [141].

### 4.2. Endogenous Protein Loading

A number of papers demonstrate CRISPR-Cas9 delivery approaches, and thus, in certain protocols, protein loading can accompany RNA loading. CRISPR-Cas possesses great potential in the therapy of cancer and hereditary and infectious diseases [142,143,144,145,146], and EVs are considered as an important carrier for CRISPR-Cas delivery. To preserve the functional activity of the packaged protein, it must be released from the vesicle membrane into recipient cells. Here, we focus on some examples of protein loading that have not been described in the previous section. Along with the transfection of donor cells, a plethora of methods has been developed to endogenously load cargo proteins into EVs (Figure 6). These methods can be divided into three distinct strategies:Fusing cargo to proteins enriched on EV membranes;Using post-translational modifications of the cargo proteins;Viral protein-assisted loading.

#### 4.2.1. Fusion with Proteins Enriched within EV Membranes

Similar to RNA packaging, proteins of interest can be fused to proteins that are enriched in EVs. For this purpose, a light-inducible loading system called exosomes for protein loading via optically reversible protein–protein interactions (EXPLOR) has been proposed [147]. Two recombinant proteins are overexpressed in this system: CIBN is conjugated to the EV-associated CD9, while the cargo protein is fused to CRY2. Similar to the RNA loading method discussed above, blue light illumination induced the dimerization of CRY2-CIBN, with subsequent enrichment of the protein of interest in EVs. In the absence of blue light, engineered cargo detaches and releases into EV intraluminal space. This method demonstrated the efficient delivery of Bax protein, super-repressor IκB protein, and Cre enzyme. However, the EXPLOR system shows limited loading efficiency with an average of 1–2 cargo molecules per particle, and although blue light did not cause cell toxicity in the study, it could still potentially be toxic to cells.

Another method is based on the use of CD63 for packaging and subsequent cleavage of the cargo into a soluble active form, releasing it into the EV lumen. A recent study employed engineered self-cleaving pH-sensitive intein to develop systems for cytosolic protein delivery, termed VSV-G plus EV-sorting Domain-Intein-Cargo (VEDIC) and VSV-G-Foldon-Intein-Cargo (VFIC) [148]. Treating reporter cells with collected particles derived from CD63-Intein-Cre and VSV-G-expressing cells leads to significant gene editing activity. To further improve the VEDIC system, the VFIC system expressed through a single vector was developed. Also, VSV-G itself has been shown to act as an efficient EV sorting domain without requiring the co-expression of an EV-sorting protein such as CD63. Furthermore, to improve VSV-G trimer formation and function, the T4 fibritin trimerization motif (foldon) was added to the construct. The efficiency of cargo delivery using the VFIC system turned out to be significantly higher (about 2–3 times) than using the VEDIC system.

Gesicles are particles generated with the help of viral proteins (VSV-G). Overexpressing VSV-G induces the generation of many VLPs with the encapsulated cargo of interest [149]. Due to the high prevalence of low-density lipoprotein (LDL) receptors, which serve as cellular receptors for vesicular stomatitis virus, gesicles have a broad tropism [150]. This method was subsequently used as the basis for a trigger-dependent system based on the interaction of FK506-binding protein (FKBP12) and FKBP12-rapamycin-binding protein (FRB) [151]. In this system, gesicles are produced for CRISPR/Cas9 RNP delivery using a system composed of four components: membrane-associated protein produced by the interaction between two domains (FKBP12 and FRB), VSV-G, Cas9, and the chosen gRNA. In the presence of rapamycin (A/C heterodimerizer), the two domains of the protein are brought together to form a complete protein that is enriched in EV membranes. Gesicle-mediated CRISPR/Cas9 RNP delivery targets the HIV long terminal repeat (LTR) reduced copy numbers of the HIV provirus and the viral protein Nef. However, VSV-G may have immunogenic properties that need to be addressed for clinical applications [152].

Another approach can be used with the specific EV population called arrestin domain-containing protein 1- (ARRDC1)-mediated microvesicles (ARMMs) [153]. ARRDC1 plays a crucial role in ARMM biogenesis at the plasma membrane. Its overexpression triggers outward membrane budding, leading to a rise in ARMM production [97]. Proteins of interest can be directly fused with ARRDC1 and incorporated into ARMMs. To deliver additional mRNA, some authors utilized the viral TAT-TAR system [153], in which ARRDC1 is fused with the TAT protein; after transfection, ARMMs are collected. The RNA cargo is loaded into EVs due to the specific interaction between the TAR RNA loop and TAT protein. In this study, ARMMs efficiently delivered functional p53 and p53 mRNA, which was translated into functional protein. However, for the packaging of larger proteins (Cas9), researchers decided to use the interaction of ARRDC1 with WW-domain-containing proteins instead of direct protein cargo fusion with ARRDC1. ARRDC1 acts as an adapter recruiting ubiquitin-protein ligases to proteins with WW-domain-containing proteins. Because of this, WW-Cas9, along with its associated sgRNA, is successfully incorporated into ARMMs [153].

In addition, some other EV-sorting proteins, like tetraspanin 14 (TSPAN14) [154], TSPAN2, and TSPAN3 [155], can be employed to load proteins into EVs. For instance, in one study, the TSPAN2 and TSPAN3 proteins outperformed the well-studied CD63 scaffold in EV-sorting ability and were utilized for robust luminal loading [155]. Furthermore, brain acid soluble protein 1 (BASP1) can be used to incorporate various proteins into the lumen of EVs, since BASP1 can bind to the inner leaflet of the plasma membrane and promote cargo sorting into EVs [156].

#### 4.2.2. Post-Translational Modifications of the Protein

Post-translational modifications (PTM) can change the properties of a newly translated protein by proteolytic cleavage or the addition of functional groups. PTM can guide proteins to be packaged inside EVs by distinct mechanisms. For instance, polyubiquitinated cargoes activate the ESCRT-dependent sorting mechanism [157], whereas myristoylated cargoes can associate with the cell membrane, which increases their encapsulation into EVs [158].

Ubiquitination [159], myristoylation [158], and palmitoylation [129] are commonly used tags for favoring the loading of proteins into EVs. Furthermore, as mentioned, N-glycosylation sites and WW tags can act as PTM sites. All these PTMs promote cargo protein incorporation, while some PTMs can also contribute to EV uptake by recipient cells and signal transduction [160]. The PTM of a protein of interest can promote its relatively efficient loading into EVs. For example, the myristoylation–palmitoylation–palmitoylation motif (MysPalm) was demonstrated to be the best method for protein loading, allowing the highest amount of Cas9 protein per EV, which was more effective than fusing the cargo protein with CD9, Rab5c, or CD81 [139].

Another study implemented a similar approach (TOP-EVs) to the above-mentioned four-component gesicle system, but added myristoylation tags to the protein [161]. Rapamycin-inducible heterodimers T82L mutant FRB (DmrC) with FKBP12 (DmrA) were used for drug-inducible dimerization. DmrC was fused to the target protein, whereas DmrA contained an N-myristoylation sequence for the enrichment of DmrA and, thus, the final protein in the cellular membrane. VSV-G was also co-transfected along with a protein of interest as an endosomal escape booster. This study shows the successful loading of GFP into TOP-EVs in the presence of rapamycin, with more than 60% of TOP-EVs containing GFP and an average of over 90 GFP molecules per vesicle. Furthermore, the efficient intraluminal loading of Cre recombinase and CRISPR/Cas9 RNP into TOP-EVs and their delivery to the cells are demonstrated.

Another possible protein-loading mechanism relies on the use of protein motifs. A recent study showed that lysosome-associated membrane protein 2, isoform A (Lamp2A) can participate in the loading of cytosolic proteins into a subpopulation of exosomes [162]. It is a novel mechanism independent of the ESCRT machinery. Proteins with amino acid sequences biochemically related to the KFERQ motif can be loaded into a specific exosome subpopulation. When a KFERQ-like motif is added to the sequence of mCherry fluorescent protein, an approximately 7-fold increase in the presence of mCherry in sEVs is observed. However, it is unclear whether the addition of a KFERQ-like motif results in protein incorporation by a distinct mechanism or if these motifs also act as potential sites for PTM. Further studies are needed to fully elucidate the loading mechanism of proteins with KFERQ-like motifs and further applicability of this method for protein packaging into EVs.

Problematically, PTMs are most likely to be cell-specific, complicating their application in protein loading. As another drawback, PTMs can be temporary and, thus, lead to protein loss before EV sorting. For instance, ubiquitinated proteins can be deubiquitinated en route to EVs [160]. Moreover, some EV isolation methods, like dUC, can alter PTMs and thus alter functionality of the cargo proteins.

#### 4.2.3. Viral Protein-Assisted Loading

Several previously mentioned systems have used VSV-G to enhance endosomal escape and cargo delivery. In this section, we will focus on some examples of viral protein-assisted loading.

Nanoblades are biological EV-mimetic nanoparticles that were originally developed from murine leukemia VLPs for the delivery of Cas9-sgRNA RNPs [163]. The approach utilizes the ability of retroviral Gag proteins to form artificial VLPs in the presence of fusogenic VSV-G and deliver them into the cytoplasm by fusion with target cells. Gag proteins (either from murine leukemia virus [MLV] or HIV) are fused with the protein of interest, with a proteolytic cleavage site designed between the desired protein and Gag. The co-expression of the wild-type Gag-Pol protein is required to release the protein of interest from Gag into the intraluminal space via protease-mediated cleavage. The treatment of cells with isolated nanoparticles (nanoblades) containing incorporated CRISPR-Cas9 systems demonstrates a more efficient and rapid induction of double-stranded DNA breaks than CRISPR-Cas loaded via transfection or electroporation. However, this method requires the co-expression of the Gag-Pol protein, which competes for space with the protein of interest, resulting in less proteins loaded per VLP [129]. Moreover, the immunogenicity of viral proteins presents a challenge for clinical use [164]. Another disadvantage is that HIV protease produced by self-cleavage of the Gag-Pol polyprotein can mediate the degradation of packaged cargo proteins.

The VSV-G protein can be exchanged to another viral protein for nanoblades’ preparation. One group used baboon retroviral envelope glycoprotein (BaEV) to transduce hematopoietic stem cells (HSCs) using nanoblades due to the lack of LDL receptor on the HSCs [163]. The obtained VLPs result in 50% genome editing when the CRISPR-Cas9 system is packaged. However, this makes nanoblades complicated to use because cells must be transfected with plasmids that encode Gag fused with the required protein, Gag-Pol protein, VSV-G, and BaEV.

Since the discovery of nanoblades, a few studies have been conducted using this method. One study confirmed the ability of HIV-derived nanoblades to incorporate and deliver cargo, along with MLV-derived nanoblades [165]. By additionally co-transfecting BaEV, the CRISPR-Cas9 system is delivered to human T cells, B cells, and hematopoietic stem and progenitor cells with no significant toxicity. In a recent study, nanoblades were used for gene knockout in organoids from murine prostate and both murine and human rectal organoids [166]. Similarly, no toxicity was seen in the organoids and no obvious off-target effects were observed after CRISPR-Cas9 knockout.

To improve the shortcomings of nanoblades, the previously mentioned ligand-induced EV packaging system termed NanoMEDIC was developed [129]. It was designed to deliver RNPs without the need for a proteolytic cleavage site, and it was based on the association of FKBP12 and FRB in the presence of a rapamycin analog. Cells stably expressing sgRNA are transfected with plasmids encoding VSV-G, Cas9-FRB, and FKBP12-Gag containing a myristoylation motif for enrichment on the plasma membrane. The effective incorporation of RNPs into VLPs is only possible in the presence of a rapamycin analog. The NanoMEDIC system shows over 90% exon skipping efficiency in skeletal muscle cells with no observed cellular toxicity. Confirming the shortcomings of the previously mentioned nanoblades and removing the fused Pol protein containing an HIV protease cleavage site results in higher editing efficiency. As expected, the same editing efficiency is demonstrated when HIV protease inhibitor Darunavir is coupled with FKBP12-Gag-Pol, compared to transfection with FKBP12-Gag. Nevertheless, Gag and HIV-1 Tat (required to drive the expression of sgRNA in producer cells) proteins can be immunogenic when used in vivo.

To sum up, although VLPs resemble EVs and can provide efficient protein loading, viral components can trigger inflammatory responses in vivo, and VLPs may have some properties that are more inherent to viral vectors than natural EVs.

### 4.3. Exogenous Loading

Exogenous loading, also known as direct loading or post-loading, is a method used for packaging therapeutics into purified EVs. Exogenous loading allows a wider range of therapeutics to be loaded into EVs, as cells are not influenced by therapeutics that may change cell viability or alter their phenotype. For instance, cell genomes do not change when a gene editing system is exogenously loaded [167].

Exogenous loading is performed using physical methods (electroporation, sonication, heat shock, freeze-thawing, transfection, or extrusion) or chemical methods (surfactant permeabilization, pH gradient modification, hypotonic dialysis, or lipid conjugation) (Figure 7).

#### 4.3.1. Physical Methods

Electroporation is commonly used due to its cost efficiency and simplicity of operation [115]. A pulsed electrical current is applied to the sample to form temporary pores in the EV membrane. Besides small molecules and nucleic acids, this method can be utilized to load protein cargo [168]. Electroporation is an efficient and fast technique, but it can induce precipitation of cargo RNA [169] and change the morphology of EV membranes with further EV aggregation [170]. Moreover, parameters like condenser capacity, voltage, number of pulses, pulse length, and interval duration should be optimized for each application [171]. To minimize EV aggregation after electroporation, the use of media containing 50 mM trehalose in PBS has been proposed [172]. Alternatively, buffers containing 400 mM sucrose in PBS can be utilized to improve loading efficiency while maintaining similar EV recovery [173].

Sonication is another physical method for exogenous cargo loading. With this method, extra mechanical shear force (sound energy) is applied to the sample and temporary pores are generated. Sonication allows high loading efficiency, but it can alter EV zeta potential and cause membrane damage and aggregation of EVs [171,174,175].

Heat shock is based on a sudden and significant increase in temperature to generate transient pores in the EV membrane. This method is relatively simple and easy to operate, does not require expensive equipment, and provides approximately the same loading efficiency as electroporation [176]. Possible drawbacks of the heat shock method include destabilization of the loaded nucleic acids, changes in the fluidity of EV membranes, and the denaturation of membrane proteins [177,178].

Freeze-thawing is also based on the temporary disruption of EV membrane integrity, which may be associated with damage caused by the formation of ice crystals [179]. Freeze-thawing is a simple method that does not require any costly equipment. Despite its advantages, this method has a relatively low drug loading efficiency [171]. During sudden temperature change, EV aggregation or destruction can occur, as well as induced protein damage [179,180].

Extrusion is another approach for exogenous cargo loading. A mixture of EVs and drugs is loaded into a syringe-based lipid extruder with 100–400 nm pores, and the EV membrane continuity decreases, resulting in cargo encapsulation within EVs [181]. Extrusion provides relatively high loading rates, improved EV size uniformity and homogeneity, does not require chemical additives or reagents, and is relatively cost-effective [115]. However, since extrusion can cause mechanical stress, it can lead to EV membrane reorganization and zeta potential changes, which can modulate immune responses and increase EV toxicity [115,171,182].

#### 4.3.2. Chemical Methods

Chemical methods are based on the application of chemical substances to increase the permeability of EV membranes for cargo loading. The effectiveness of chemical methods is highly dependent on the chemical structure of the desired cargo.

EVs can be loaded with nucleic acids in a similar manner to cell transfection. Transfection with Exo-Fect (a cationic transfection reagent) is the most efficient method for miRNA loading, compared to electroporation, heat shock, saponin-based loading, and the modification of miRNA with cholesterol [183]. This efficiency is observed for EVs isolated from various sources. The polyvalent cationic transfection reagent forms a complex with nucleic acids that facilitates loading via charge–charge interactions between the complex and EVs. EV transfection does not depend on cargo concentration and, therefore, improved loading is anticipated [184]. Nevertheless, transfection can affect EV mean size and zeta potential [175] and cause problems such as cytotoxicity. Moreover, difficulties in eliminating transfection reagent residue still remain; as aforementioned, lipid micelles from transfection reagents can mimic cargo-loaded EVs [185].

Chemical permeabilization mainly utilizes saponin, a surfactant molecule that can complex with cholesterol in cell membranes, generating pores [186]. Permeabilization with saponin is a relatively cheap and simple technique that provides a high loading efficiency [187]. Nowadays, chemical permeabilization is rarely used due to saponin’s toxicity and hemolytic activity in vivo [174]. Saponin can damage EV membranes and change EV zeta potential, and not all cargo molecules can be mixed with surfactants [115,175].

pH gradient modification was originally developed to encapsulate weakly basic drugs (e.g., doxorubicin [DOX]) within liposomes [188]. When the drug is incubated at a neutral pH with particles that have an interior acidic core, it diffuses into the lumen of the particles. While a pH gradient is more effective for entrapping basic drugs, a high transmembrane pH gradient (pH~2) can still be applied for negatively charged molecules. Thus, a pH gradient across EV membranes can be used to load nucleic acids into EVs. To achieve this, one group dehydrated EVs in 70% ethanol and then rehydrated them in an acidic citrate buffer (pH = 2.5) to establish the pH gradient [189]. This approach allowed the enhanced loading of miRNA, siRNA, and single-stranded DNA into EVs with about the same loading efficiency as that achieved by sonication, electroporation, or chemical permeabilization. Simultaneously, size and zeta potential were unaffected by this method [171]. However, the addition of ethanol causes EV protein degradation and changes EV stability [167]; for example, milk-derived EVs are unstable during the ethanol dehydration step [190]. Moreover, a low pH may induce the acid hydrolysis of lipids like phosphatidylcholine, enhancing drug leakage and particle instability [191].

Hypotonic dialysis is a method for loading cargo during EV swelling. A hypotonic solution is added to form temporary pores in EVs, allowing water-soluble molecules to move into the vesicles with the water flow. Then, EVs are dispersed in an isotonic solution, which favors EVs transition to their normal state. Hypotonic dialysis is a simple technique that provides high efficiency and does not require any specialized equipment, but it can alter the size and charge of EVs and cause the degradation of protein and peptide cargo [171,192]. Furthermore, one study showed a relatively low EV uptake by the cells after cargo loading with this method [187].

Lipid conjugation may also be used to improve the loading of therapeutic RNAs and other hydrophilic ligands (e.g., peptides) into EVs. The covalent conjugation of RNA with lipids is mostly utilized by cholesterol conjugation, but fatty acids, sterols, and vitamins can also be anchored to the RNA of interest. For instance, conjugation with docosanoic acid and α-tocopheryl succinate shows the same or better siRNA loading into EVs than conjugation with cholesterol [193]. Vitamin E-conjugated siRNA supports the best loading into EVs, outperforming cholesterol-conjugated RNA. Hydrophobicity drives the integration of lipid-conjugated hydrophilic molecules into the EV membrane during simple co-incubation. Synthesized hydrophobic RNAs package into EVs more efficiently [193]. Despite the advantages of lipid-conjugated RNA, most RNA attaches to the EV surface instead of being incorporated [194]: the reduced zeta potential and increased EV size are noted after loading, indicating the deposition of RNA on the EV surface [195]. Furthermore, hydrophilic cholesterol-anchored ligands are shed from EVs in the presence of serum in less than two hours [196], potentially explaining the generally low efficiency of the silencing activity of cholesterol-conjugated siRNA transferred by EVs in vivo.

## 5. Challenges of Loading Cargo into EVs

Although numerous loading techniques for therapeutic molecules have been developed, they still suffer from an inadequate efficiency inferior to that of lipid-based NPs [197,198]. Large hydrophilic molecules are particularly difficult to package due to the presence of EV lipid bilayer membranes that protect naturally incorporated material.

Endogenous loading provides a continuous production of drug-loaded EVs without damaging EV membranes, but most pre-loading methods are time-consuming and laborious. Compared to exogenous loading, endogenous packaging results in low loading efficiency for smaller molecules [112,192]. Moreover, it is hard to control the amount of packaged cargo into EVs with exogenous loading, and this amount depends on transfection efficiency and the viability of the producing cells. As another drawback, loaded molecules may influence EV-producing cells, impacting their physiology and, therefore, changing the constitution of released EVs and/or limiting EV production [199].

Exogenous loading has its own disadvantages. EVs have plenty of proteins and nucleic acids inside [200]. As mentioned, most exogenous packaging methods can disrupt EV integrity, thereby altering the payload and delivery vehicle [161]. Furthermore, exogenous loading shows a lower loading efficiency for larger molecules, such as proteins and nucleic acids larger than 1000 bp [201].

Thus, each type of cargo requires special loading methods. In the case of small molecules, the application of endogenous methods results in low efficiency and possibly affects the viability or metabolism of EV-producing cells. At the same time, the application of novel exogenous protocols based on ion gradients allows for a high drug loading efficiency by utilizing active transport mechanisms, achieving up to 60% loading efficiency for doxorubicin [198]. The packaging of proteins with exogenous methods is complicated or can result in EV damaging (for example, during electroporation). The use of endogenous methods, such as viral-assisted loading, or the use of dimerization domains induce the effective packaging of desired proteins. For example, the use of the nanoblades system for large Cas9 protein loading results in ~60 Cas9 proteins per vesicle [166]. In the context of RNA packaging, genetically engineered constructs enable efficient loading of RNA; however, they necessitate the introduction of substantial amounts of endogenous components. The chemical transfection of isolated EVs with cargo RNA, resulting in the formation of hybrid vesicles, presents a cost-effective alternative for small RNA molecules. Further experiments are needed to identify the most effective protocols for packaging various types of cargo.

## 6. Surface Display of Functional Moieties on EVs

Fusion with EV-specific proteins is the most commonly used approach to display functional moieties on the surface of EVs. Displayed proteins or peptides can be used as targeting molecules, receptor agonists or antagonists, etc. Proteins or peptides can be included in extracellular loops of tetraspanins (CD63, CD9, and CD81) or fused to the extracellular domains of proteins highly enriched in EVs, such as lysosome-associated membrane protein 2, isoform B (Lamp2B), or lactadherin. Moreover, functional moieties can be displayed on the surface of EVs by employing transmembrane proteins enriched in EVs, such as prostaglandin F2 receptor inhibitor (PTGFRN), TSPAN14, TSPAN2, and TSPAN3.

Lamp2b is the most frequently used protein for displaying targeting moieties through cell engineering. For instance, Lamp2 fused to the neuron-specific rabies viral glycoprotein (RVG) peptide delivered EVs containing siRNA targeting GAPDH specifically to neurons, microglia, and oligodendrocytes in a mouse model [202]. Upon intravenous administration of these decorated RVG-modified EVs, a strong neuron-specific knockdown of GAPDH gene expression was observed.

Lactadherin is a secretory protein that specifically binds to the EV surface through phosphatidylserine and becomes a membrane-associated protein [203]. Fusing anti-HER2 single-chain variable fragments with the C1C2 domain of lactadherin protein demonstrates their specific delivery to HER2-positive human breast tumor xenografts [204]. In another study of in vivo tracing of EVs, a fusion protein was designed consisting of *Gaussia* luciferase (gLuc), C1C2 domains of lactadherin, and the N-terminal secretion signal of lactadherin [205]. This protein resulted in a strong luciferase activity in the EV-containing fraction.

Functional moieties also can be included in the large extracellular loop (2nd loop) of CD63 for surface display. For example, the large extracellular loop of CD63 was genetically engineered for the surface display of the albumin-binding domain to increase EV circulation time [206]. However, the surface display of large molecules can compromise stability and the EV-sorting ability of the fusion protein [207]. Alternatively, targeting molecules can be fused with specific CP05 peptides that bind specifically to CD63 molecules at the EV surface and expose the targeting molecule. For example, the functionalizing of EVs with a muscle-targeting peptide and EV loading with a splice-correcting oligomer for dystrophin exon skipping induced functional rescue and the formation of dystrophin-positive muscle fibers in mdx mice [208].

Alternative technology for ligand display include anchoring via lipid molecules. In this technology, the targeting molecule is fused with a lipid (for example, cholesterol or phospholipid). The incubation of construct with the EV results in the incorporation of the lipid molecule into the EV lipid membrane and the surface display of the targeting ligand [209,210,211]. For example, Pi et al. used cholesterol-conjugated arrow-shaped RNA modified with PSMA-specific or EGFR-specific RNA aptamers or folate molecules for targeting in prostate cancer, breast cancer, and colorectal cancer cells in mice models, respectively [210].

Another approach was demonstrated using a glycophosphatidylinositol (GPI) anchor signaling peptide derived from a decay-accelerating factor (DAF) fused to an EG1a nanobody with a high affinity to EGFR. As a result, targeting nanobodies were enriched in EVs and exposed at the EV surface leading to an increased uptake by EGFR-overexpressing cells compared to unfunctionalized vesicles [212].

While most studies for EV surface display have used protein and peptide ligands, glycan ligands can also be used for cell-specific targeting. This is accomplished by co-expressing a glycosylation domain inserted into the large extracellular loop of CD63 and fucosyltransferase VII (FUT7) or IX (FUT9) responsible for the surface display of the glycan of interest [207]. FUT7 and FUT9 catalyze the fucosylation of one of the residues of a short glycosylation domain. These structures act as glycan ligands and mediate adhesion to activated endothelial cells or dendritic cells, respectively. Various available glycosyltransferases can be used for the surface display of glycans with different biological functions, making this approach attractive.

The exposure of targeting molecules independently of surface proteins can be performed by a click chemistry approach. In this way, the exact chemical groups of all surface proteins interact with reactogenic chemical compounds in mild conditions, forming an activated chemical group. During the second step, targeting molecules react with the activated group by forming covalent links, resulting in the exposure of the targeting molecule. Reactions can be performed in neutral physiological buffers without catalysts. MSC-derived EVs modified with the c(RGDyK) peptide and loaded with anti-inflammatory drug curcumin were tested in cerebral artery occlusion in mice models. The functionalized particles efficiently bound to target integrin αvβ3 overexpressed in ischemic brain sites, and they reduced inflammation and cells apoptosis at the site of ischemia [213].

Displayed molecules on the EV surface may be unstable in vivo and can be attacked by membrane proteases [214]. Another challenge is the efficient display of functional moieties on the surface of EVs. For instance, some peptides fused to Lamp2b were not displayed efficiently on the EV surface [215]. Moreover, EV markers are not found on all EVs, or are not present at densities optimal for engineering [216]. Overall, the efficient surface exposure of desired molecules is an area requiring further research.

## 7. Extracellular Vesicles in Clinical Practice

Several other researches have previously reviewed the landscape of clinical trials related to EVs and exosomes [96,217]. In particular, Ghodasara et al. have identified more than 40 clinical trials of EV-based therapeutics [96]. In our review, we have conducted a search on the clinicaltrials.gov website and we were able to retrieve 221 interventional clinical trial records by the query “exosome” and 136 interventional clinical trial records by the query “extracellular vesicle”. Due to the scope of our review, we have made a decision to exclude all of the trials in regard to EV biomarkers and diagnostics, which resulted in a total of 99 unique interventional clinical trials. In total, 59 out of 99 (59.5%) trials were conducted with the use of MSC-derived extracellular vesicles. In eight records, blood and blood components-derived EVs were utilized. Among other EV sources present in the studies, antigen-presenting cells, induced pluripotent stem cells, T-cells, and plant cells were present. In the majority of the studies, EVs were utilized without any surface modification or cargo loading, and only a few trials utilized engineered exosomes. However, we have found several records regarding therapeutics based on CD24-overexpressing cells-derived exosomes (NCT05947747, NCT04969172, NCT04902183, and NCT04747574). CD24-enriched exosomes were manufactured with the help of the transfection of Expi293F cells with plasmid-encoding CD24. Unfortunately, there are no results for these clinical trials available. Previous publications demonstrated that CD24-enriched exosomes can mitigate ARDS in murine models. In contrast to immune suppression, caused by dexamethasone, CD24-enriched exosomes did not suppress the immune system to the levels below the baseline [218]. The clinical trials of EV products are summarized at the Appendix A.

## 8. Conclusions

There are currently no FDA-approved EV products, and many studies are still in the early stages, indicating that EV-based therapeutics have a long way to go before they can be widely used in clinical therapy. Moreover, issues like the short half-life of EVs and cell type-specific targeting remain, and they can hinder EV use [219]. Despite the promising applications of EVs, major limitations linger in our current understanding of EV biogenesis and cargo encapsulation. The understanding of this processes will allow us to extremely improve the packaging of therapeutic cargo to improve the efficiency of delivery by extracellular vesicles for most perspective drugs, for example, CRISPR-Cas gene editing systems. For example, the efficiency of CRISPR-Cas packaging in the EV-mimetic nanoblades system could achieve tens of Cas-protein copies per vesicle that result in superior gene editing efficiency [163]. For now, EVs are considered as the optimal delivery vehicles for CRISPR-Cas systems of any type. The targeted delivery of small-molecular chemotherapy drugs into tumors is a second potential area for EV-based delivery. It was previously demonstrated that EV loaded with cytotoxic compounds demonstrates superior cytotoxicity compared to chemotherapeutic alone [220]. Additionally, several studies have demonstrated that extracellular vesicles (EVs) can overcome certain drug resistance mechanisms, particularly in cells that overexpress P-glycoprotein during drug delivery [221]. Optimized delivery methods will enhance the accumulation of cytotoxic drugs in cancer cells, including metastases, while simultaneously reducing common drug toxicity. The intrinsic ability of MSC-derived EVs to target inflamed tissues facilitates the delivery of small-molecule antioxidants and protective compounds to ischemic sites in patients with myocardial infarctions and strokes, thereby reducing the area of necrotic lesions and promoting regeneration. Furthermore, EVs have the potential to replace liposomes in certain applications of RNA delivery, and clinical trials investigating the use of EVs for the delivery of RNA-interfering molecules are currently underway [222]. Improvement in the surface display of targeting ligands will open a way to a highly specific biodistribution of vesicles into the target tissue, resulting in further increase in drug delivery by EVs.

Overall, extracellular vesicles represent a promising platform for drug delivery. In the context of cargo delivery, EVs offer nearly unmatched biocompatibility. They are characterized by prolonged circulation times, low toxicity, and reduced immunogenicity compared to other nanoplatforms, such as liposomes [223]. EVs have also demonstrated an mRNA delivery efficiency superior to that of lipid nanoparticles in primary human cells. The intramuscular injection of EVs carrying mRNA encoding SARS-CoV2 spike protein resulted in efficient mRNA expression near the injection site, and resulted in a sustained immune response to the viral protein. Additionally, no attenuation of mRNA expression was observed after the re-administration of the exosomes [224]. Similar results were shown in another study in an in vivo model. The EV-based siRNA delivery system was able to outperform lipid nanoparticles for breast cancer siRNA delivery [225]. Apart from the great biocompatibility and delivery efficiency, some types of EVs possess inherent targeting capabilities. For example, exosomes, derived from tumor cells, are known to homotypically target tumor cells of the same type [226]. Additionally, the surface of EVs can be further modified with targeting and auxiliary molecules, which enhances the efficiency of EV-mediated delivery. In several studies, EVs decorated with “don’t eat me” signals have demonstrated prolonged circulation in the bloodstream and reduced clearance by macrophages [227,228]. While decoration with targeting peptides results in the targeted delivery of the cargo [228,229], the development of a cost-effective delivery system will require us to combine several approaches during manufacturing. Cell lines that express EV secretion-boosting molecules, packaging systems, and targeting molecules will give us the opportunity to achieve an effective therapeutic effect with acceptable drug costs. In summary, EVs have the potential to become the “gold standard” of drug delivery vehicles in the future.

## Figures and Tables

**Figure 1 ijms-25-10401-f001:**
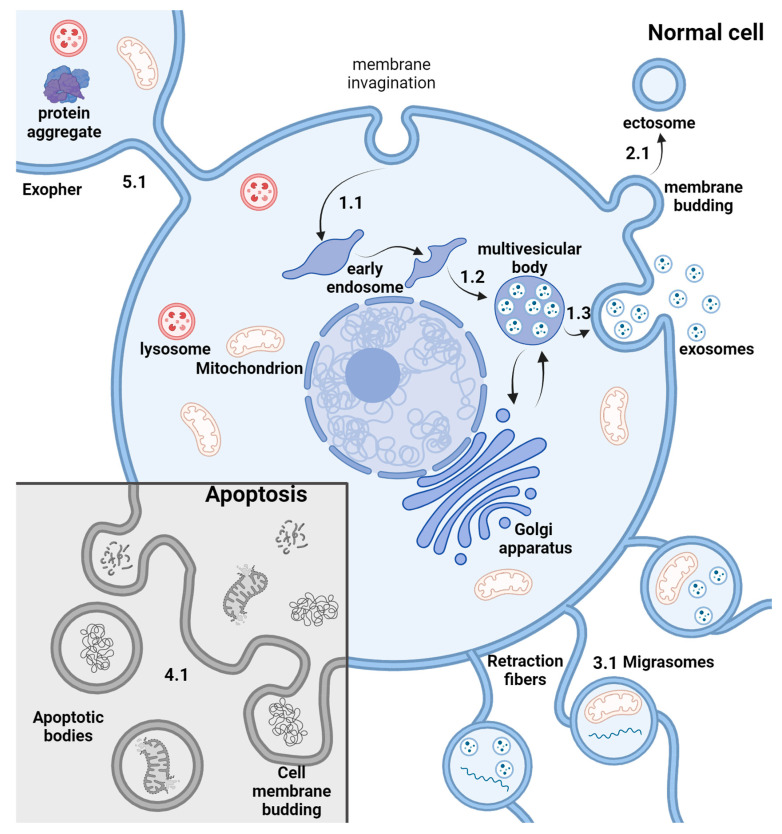
The biogenesis of extracellular vesicles. Exosome biogenesis: 1.1 Early endosome forms from cell membrane invagination. 1.2 During the maturation process, membrane invaginations form on endosomal membrane, which subsequently bud into endosomal lumen, becoming intraluminal vesicles. 1.3 The multivesicular body fuses with the cellular membrane, releasing exosomes into the extracellular space. Ectosome biogenesis: 2.1 Ectosome biogenesis happens through direct membrane budding and fission. Migrasome biogenesis: 3.1 Migrasomes form on retraction fibers during cellular migration due to interactions with the extracellular matrix. Apoptotic bodies biogenesis: 4.1 Apoptotic bodies from the fragmented cellular membrane during apoptosis. Exopher biogenesis: Exopher forms by cell membrane budding. For a period of time it remains connected to the cell with a nanotube, but it is subsequently disconnected from the cell. 1.1 Early endosomes are formed through the invagination of the cell membrane. 1.2 During the maturation process, membrane invaginations occur on the endosomal membrane, which subsequently bud into the endosomal lumen, resulting in the formation of intraluminal vesicles. 1.3 Multivesicular bodies fuse with the cellular membrane, releasing exosomes into the extracellular space. Ectosome Biogenesis: 2.1 Ectosomes are generated through direct membrane budding and fission. Migrasome Biogenesis: 3.1 Migrasomes form on retraction fibers during cellular migration as a result of interactions with the extracellular matrix. Apoptotic Body Biogenesis: 4.1 Apoptotic bodies arise from the fragmentation of the cellular membrane during apoptosis. Exopher Biogenesis: Exophers are formed by the budding of the cell membrane. For a period, they remain connected to the cell via a nanotube but are subsequently detached from the cell.

**Figure 2 ijms-25-10401-f002:**
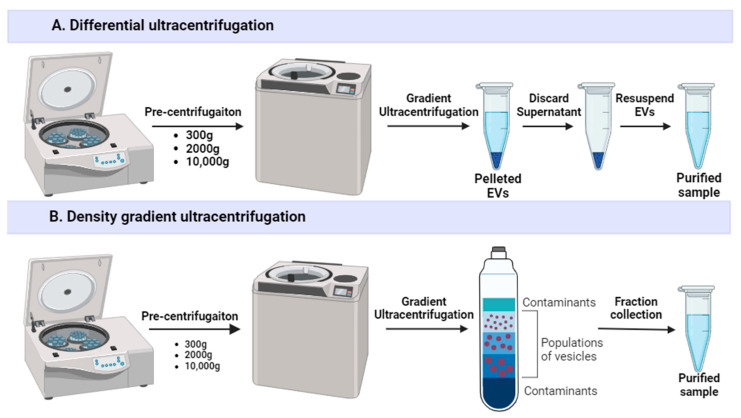
Ultracentrifugation-based EV purification methods. (**A**) Differential ultracentrifugation begins with several pre-centrifugation steps to remove cells, cell debris, and large non-exosomal vesicles. Subsequently, extracellular vesicles (EVs) are pelleted using an ultracentrifuge. The supernatant, which contains contaminants, is discarded, and the EVs are resuspended in an appropriate buffer. (**B**) Density gradient ultracentrifugation also involves multiple pre-centrifugation steps to eliminate large contaminants. Following pre-centrifugation, the sample is applied to the density gradient, and ultracentrifugation is conducted. The fraction containing the desired EV population is then collected.

**Figure 3 ijms-25-10401-f003:**
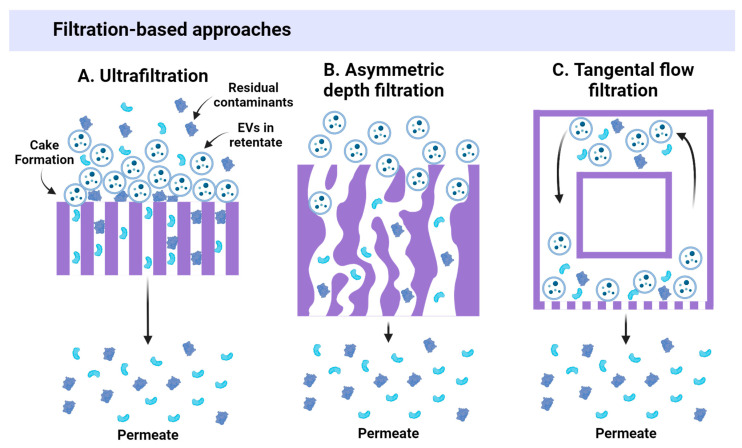
Ultrafiltration-based EV purification methods. (**A**) During ultrafiltration, pressure forces particles through the filter membrane. Smaller particles permeate the membrane and are discarded, while EVs cannot permeate the membrane and remain in the retentate. However, the inability of larger particles to permeate the membrane leads to the formation of a cake, which hinders the passage of residual smaller molecules through the membrane, resulting in reduced purification efficiency. (**B**) During asymmetric depth filtration, pressure forces the sample through a filter membrane characterized by asymmetric irregular pores. These irregular pores allow particles to enter the membrane, but their movement is impeded based on size. Smaller particles can move further or even pass through the membrane, while larger particles become entrapped and can be eluted later. (**C**) Tangential flow filtration is based on the continuous cyclic flow of particles through a closed system that incorporates a filter membrane into the wall of a filter cartridge. The pressure in the system drives the sample through the membrane, while the feed flow prevents cake formation.

**Figure 4 ijms-25-10401-f004:**
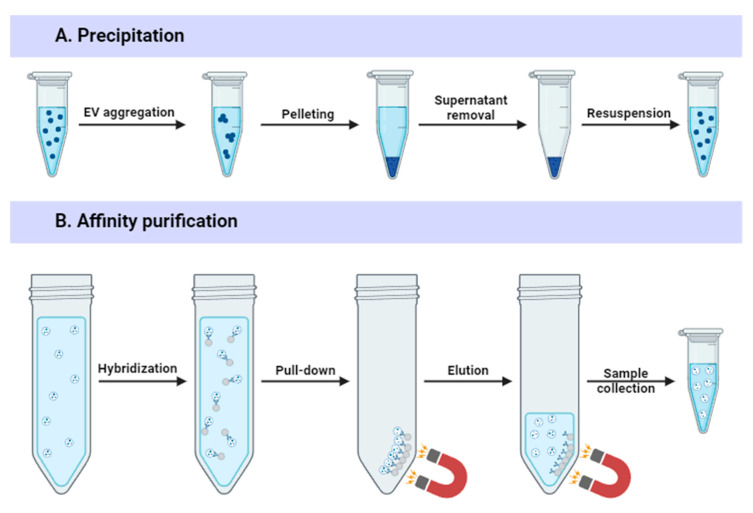
Precipitation and affinity-based EV purification methods. (**A**) The addition of the cationic polymer induces EV aggregation. The sample is centrifuged, which results in EV pelleting and some of the contaminants residing in the supernatant. The supernatant is subsequently removed and EVs are resuspended in a suitable buffer. (**B**) During affinity purification, antibodies against EV markers conjugated to the magnetic beads are added to the sample. After the binding of antibodies, the beads are pulled down by magnetic force. The supernatant is replaced by the elution buffer, resulting in the release of the EVs. Afterward, the sample is collected.

**Figure 5 ijms-25-10401-f005:**
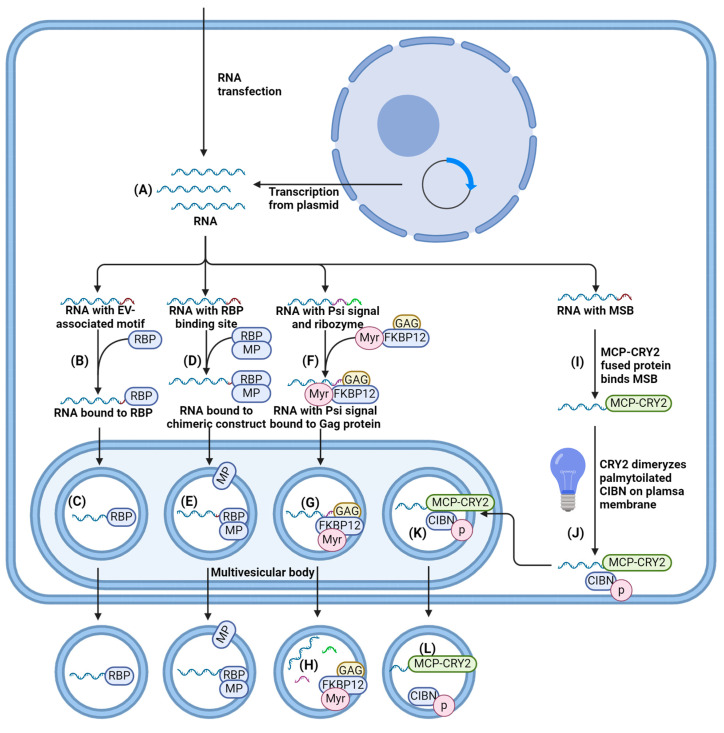
Endogenous RNA loading methods. (**A**) RNA is delivered into the cell via transfection with RNA or a plasmid encoding the RNA. (**B**) The EV-associated motif in RNA facilitates the binding of EV-associated RNA-binding proteins (RBPs). (**C**) EV-associated RBPs assist in the sorting of RNA into extracellular vesicles (EVs). (**D**) A chimeric construct consisting of an RBP and an EV membrane protein (MP) binds to the RBP-binding site in RNA. (**E**) The RBP-MP complex bound to RNA is sorted to the EV membrane. (**F**) A chimeric protein consisting of myristoylated FKBP12 and GAG binds to the Psi signal in RNA. (**G**) Due to myristoylation, the RNA-bound construct is sorted into EVs. (**H**) A ribozyme encoded in the RNA induces self-cleavage and the release of the RNA. (**I**) MCP fused to CRY2 binds to the MSB site in the RNA. (**J**) Blue light induces the dimerization of CRY2 with palmitoylated CIBN, which is enriched on the cell membrane. (**K**) The dimer bound to RNA is sorted into the EV. (**L**) In the absence of blue light, the dimer disassembles.

**Figure 6 ijms-25-10401-f006:**
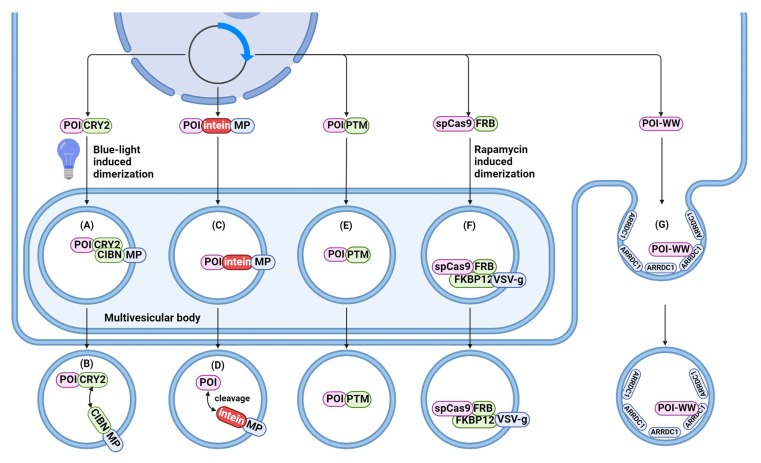
Endogenous protein loading methods. (**A**) The protein of interest (POI) is fused with CRY2. Upon exposure to blue light, the POI-CRY2 complex dimerizes with CIBN, which is fused to an EV membrane protein (MP). (**B**) In the absence of blue light, the dimer disassembles, releasing the protein into the EV lumen. (**C**) The chimeric protein, consisting of the POI, a self-cleaving intein, and the MP, is sorted into the EV. (**D**) The intein cleaves the construct, releasing the POI into the EV lumen. (**E**) Post-translational modifications (PTMs), such as myristoylation or palmitoylation, facilitate the sorting of the protein into EVs. (**F**) Rapamycin induces the dimerization of SpCas9-FRB with FKBP-VSV-g. Due to VSV-g, the construct is sorted into the EV. (**G**) The POI, fused to a WW signal, binds to ARRDC1, leading to their sorting into ARRDC1-mediated microvesicles.

**Figure 7 ijms-25-10401-f007:**
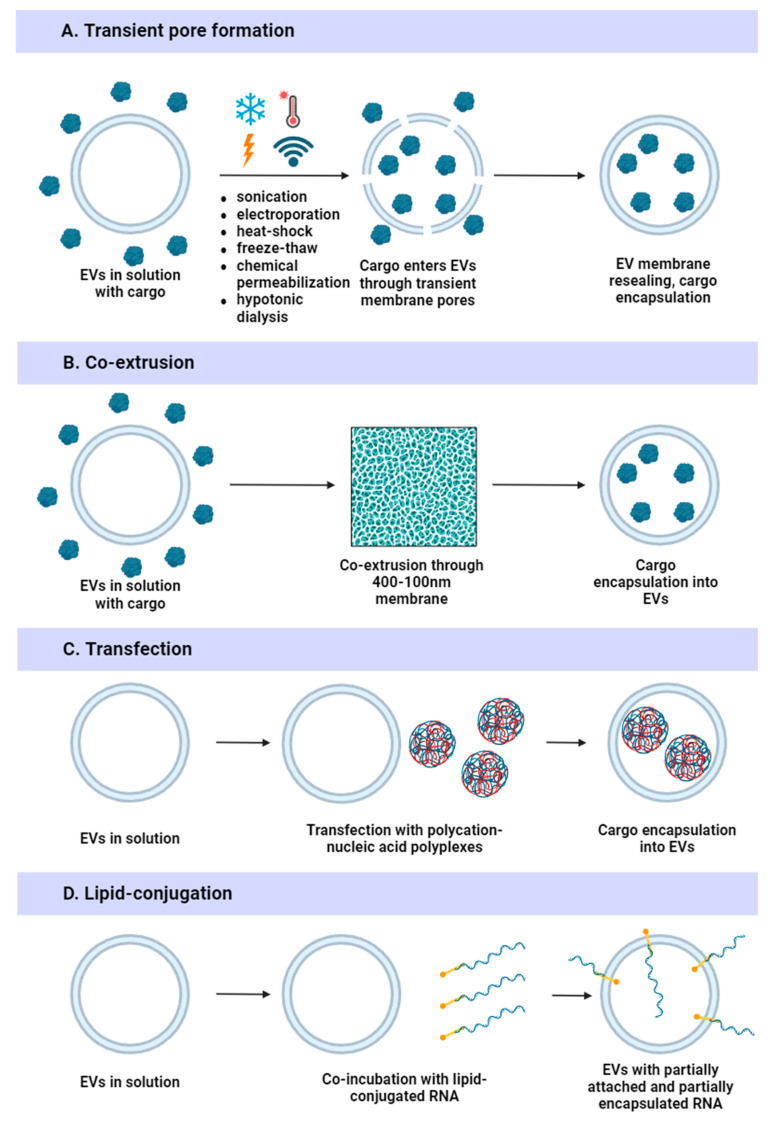
Exogenous cargo loading methods. (**A**) Several methods, including sonication, electroporation, freeze-thaw cycles, heat shock, chemical permeabilization, and hypotonic dialysis, induce transient pore formation in membranes. Through these pores, cargo from the extravesicular space can enter the vesicle. Subsequently, the membrane reseals, trapping the cargo inside. (**B**) Co-extrusion leads to a transient decrease in membrane continuity, facilitating the entry of cargo into EVs. (**C**) Similar to cells, EVs can be transfected with nucleic acids using cationic transfection reagents. (**D**) Cargo can be conjugated to lipids and subsequently incorporated into the EV membrane; however, the cargo may exist both within the vesicle and on its surface.

## Data Availability

No data were generated in this manuscript.

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
