# Peer review of "Basic Guide for Approaching Drug Delivery with Extracellular Vesicles"

_ijms, 2024, doi:10.3390/ijms251910401_

Round 1
Reviewer 1 Report
Comments and Suggestions for Authors
In this review, the authors provide discussion of methods to enhance the isolation and purification of EVs, strategies for improving cargo packaging—including proteins, RNAs, and small-molecule drugs—and technologies for surface display of targeting ligands to enhance EV targeting. While the review is comprehensive, it should be noted that several similar reviews have been published recently in this area. To strengthen their work, the authors are encouraged to clearly define their unique contributions and highlight what sets this review apart from others. Additionally, the clinical relevance of the content is relatively weak and needs further elaboration. Therefore, a major revision is recommended. Specific comments are as follows:
1. In the introduction, it is recommended to define EVs in line with MISEV guidelines.
2. In Title 2, using "EV biogeneration" or "biogenesis'; would be more appropriate.
3. Regarding the table, please consider including the recovery rate for each isolation/purification method.
4. The section on low EV yield is limited in scope. It would be beneficial to also include other scale-up methods, such as electrical stimulation (doi.org/10.1002/advs.202302622) and biological transfection (doi.org/10.1038/s41467-018-03733-8). These approaches reduce cell damage while improving EV quality (e.g., increased cargo loading) and should be discussed.
5. Section 4.1.1, some physical and transient transfection example should be included in the endogenous loading, for example, using nanoeletroporation for realizing diverse cargos loading in EV. The author should review those related work in this article
6. The clinical aspects need further attention. Please elaborate on the current status of clinical applications in the EV field, incorporating more recent EV review papers related to clinical translation (e.g., doi.org/10.1002/adhm.202301010; 10.1016/j.tibtech.2024.08.007). A table for recent clinical cases is encouraged.
Comments on the Quality of English Languagena
Author Response
Reviewer 1:
- In the introduction, it is recommended to define EVs in line with MISEV guidelines.
Response 1: We agree. EVs are now defined according to MISEV guidelines. The following text was added: “Extracellular vesicles (EVs) according to MISEV2023 are “Particles that are re-leased from cells, delimited by a lipid bilayer, that cannot replicate on their own”.
- In Title 2, using "EV biogeneration" or "biogenesis'; would be more appropriate.
Response 2: Thank you, the title 2 was corrected.
- Regarding the table, please consider including the recovery rate for each isolation/purification method.
Response 3: Thank you for useful suggestion. Table 1 was supplemented with data on the recovery rates of these methods.
- The section on low EV yield is limited in scope. It would be beneficial to also include other scale-up methods, such as electrical stimulation (doi.org/10.1002/advs.202302622) and biological transfection (doi.org/10.1038/s41467-018-03733-8). These approaches reduce cell damage while improving EV quality (e.g., increased cargo loading) and should be discussed.
Response 4: suggested papers are now discussed in different parts of the manuscript. The following new text was added: “Alternative approaches to enhancing extracellular vesicle (EV) yield include electrical stimulation and biological transfection. These methods enable the scaling up of EV production while minimizing severe damage to producer cells by stimulating the ex-pression of genes that influence EV production. Cellular nanoelectroporation [75] and biological transfection [76] can also provide high endogenous loading of cargo such as functional mRNA.”
“Physical transfection, called electrical stimulation, can also be used for endogenous loading of diverse cargoes. For example, Y. Ma et al., used nanoelectroporation to load human bone morphogenetic protein 2 (BMP-2) and human vascular endothelial growth factor A (VEGF-A) mRNAs into EVs. Nanoelectroporation principle is based on seed-ing of cells onto track-etched membrane with nanopores (400 nm), and loading of PBS supplemented with desired plasmids under the membrane. Application of electric fields across the membrane induce transfer of plasmid into cells via nanopores, result-ing in efficient transfection and relatively low toxicity [75]. Similar principle was used to load EVs secreted by dermal fibroblasts with COL1A1 RNA for treatment of pro-toaged skin [120]. Nevertheless, scalability of method requires additional investigation.”
“Additionally, transfection of EV producing cells with “booster” plasmid results in evident increase of EV secretion (up to 15-40-fold).”
- Section 4.1.1, some physical and transient transfection example should be included in the endogenous loading, for example, using nanoeletroporation for realizing diverse cargos loading in EV. The author should review those related work in this article.
Response 5: Thank you for this suggestion. New text was added into the manuscript: “Physical transfection, called electrical stimulation, can also be used for endogenous loading of diverse cargoes. For example, Y. Ma et al., used nanoelectroporation to load human bone morphogenetic protein 2 (BMP-2) and human vascular endothelial growth factor A (VEGF-A) mRNAs into EVs. Nanoelectroporation principle is based on seed-ing of cells onto track-etched membrane with nanopores (400 nm), and loading of PBS supplemented with desired plasmids under the membrane. Application of electric fields across the membrane induce transfer of plasmid into cells via nanopores, result-ing in efficient transfection and relatively low toxicity [75]. Similar principle was used to load EVs secreted by dermal fibroblasts with COL1A1 RNA for treatment of pro-toaged skin [120]. Nevertheless, scalability of method requires additional investigation.”
- The clinical aspects need further attention. Please elaborate on the current status of clinical applications in the EV field, incorporating more recent EV review papers related to clinical translation (e.g., doi.org/10.1002/adhm.202301010; 10.1016/j.tibtech.2024.08.007). A table for recent clinical cases is encouraged.
Response 6: We agree that discussion of current clinical trials of EVs will evidently improve the manuscript. Clinical trials of EV products are now listed in Table S1. Discussion of trials based on EV additionally loaded with therapeutic molecules was added as the following text: “Several other researches have previously reviewed the landscape of clinical trials related to EVs and exosomes [96,217]. In particular, Ghodasara et al. have identified more than 40 clinical trials of EV-based therapeutics [96]. In our review we have con-ducted search on clinicaltrials.gov website and were able to retrieve 221 interventional clinical trial records by the query “exosome” and 136 interventional clinical trial rec-ords by the query “extracellular vesicle”. Due to the scope of our review we have made a decision to exclude all of the trials in regards to EV-biomarkers and diagnostics, which resulted in total of 99 unique interventional clinical trials. 59 out of 99 (59,5%) trials were conducted with the use of MSC-derived extracellular vesicles. In 8 records blood and blood components-derived EVs were utilized. Among other EV sources present in the studies, antigen-presenting cells, induced pluripotent stem cells, T-cells and plant cells were present. In majority of the studies, EVs were utilized without any surface modification or cargo loading, and only a few trials utilized engineered exo-somes. However, we have found several records regarding therapeutics based on CD24-overexpressing cells-derived exosomes (NCT05947747, NCT04969172, NCT04902183, NCT04747574). CD24-enriched exosomes were manufactured with help of transfection of Expi293F cells with plasmid encoding CD24. Unfortunately, there are no results for these clinical trials available. Previous publications demonstrated that CD24 enriched exosomes can mitigate ARDS in murine model. In contrast to immune suppression, caused by dexamethasone, CD24-enriched exosomes did not suppress the immune system to the levels below the baseline [218]. Clinical trials of EV products are summarized at the Table S1.”
Reviewer 2 Report
Comments and Suggestions for Authors
In this review, methods to improve the isolation and purification of Extracellular vesicles (EVs), approaches to enhance cargo packaging—including proteins, RNAs, and small-molecule drugs—and technologies for displaying targeting ligands on the surface of EVs to facilitate improved targeting were discussed in great detail. This guide can be applied to the development of novel classes of EV-based therapeutics and to overcoming existing technological challenges.
The review is well-organized and well-written. It can be considered for publication after addressing the following minor concerns:
a) In the introduction section, the authors should highlight how the current review is different from the ones published on Extracellular vesicles. Has the same information been compiled elsewhere? A clear statement should be made regarding the novelty of the content of the review.
b) The authors are advised to provide a more give detailed perspective and future of the EV systems mentioned in the review article in the conclusion section.
Author Response
Reviewer 2:
1) In the introduction section, the authors should highlight how the current review is different from the ones published on Extracellular vesicles. Has the same information been compiled elsewhere? A clear statement should be made regarding the novelty of the content of the review.
Response 1: Thank you for the suggestion. New text is now added into the Introduction section: “Several previous reviews extensively described different aspects of EV biogenesis, production and purification for fabricating formulations with immune-regulatory and pro-regenerative properties [12–14]. In this review, we focus on providing an up-to-date, optimized technical workflow for developing EV-based drug formulations and summarize methods for the production and purification of EVs, as well as tech-niques for cargo loading. Additionally, we will discuss the primary approaches for the presentation of targeting ligands in EV-based drug delivery systems and describe cur-rent status of clinical trials of EV products containing loaded therapeutic cargo.”
2) The authors are advised to provide a more give detailed perspective and future of the EV systems mentioned in the review article in the conclusion section. [Плюс-минус накидал, остались ссылки]
Response 2: Thank you. This discussion is now added into the Discussion section, as follows:
“For now, EV are considered as optimal delivery vehicles for CRISPR-Cas systems of any types. Targeted delivery of small-molecular chemotherapy drugs into tumors is a second potential area for EV-based delivery. It was previously demonstrated that EV loaded with cytotoxic compounds demonstrate superior cytotoxicity compared to chemotherapeutic alone [220]. Additionally, several studies have demonstrated that extracellular vesicles (EVs) can overcome certain drug resistance mechanisms, particu-larly in cells that overexpress P-glycoprotein during drug delivery [221]. Optimized de-livery methods will enhance the accumulation of cytotoxic drugs in cancer cells, in-cluding metastases, while simultaneously reducing common drug toxicity. The intrin-sic ability of MSC-derived e EVs to target inflamed tissues facilitates the delivery of small-molecule antioxidants and protective compounds to ischemic sites in patients with myocardial infarctions and strokes, thereby reducing the area of necrotic lesions and promoting regeneration. Furthermore, EVs have the potential to replace liposomes in certain applications of RNA delivery, and clinical trials investigating the use of EVs for the delivery of RNA-interfering molecules are currently underway [222]. Improve-ment in surface display of targeting ligands will open a way to highly specific biodis-tribution of vesicles into target tissue, resulting in further increase of drug delivery by EVs. Overall, extracellu-lar vesicles represent a promising platform for drug delivery. In the context of cargo delivery, EVs offer nearly unmatched biocompatibility. They are characterized by pro-longed circulation times, low toxicity, and reduced immunogenicity compared to other nanoplatforms, such as liposomes [223]. EVs have also demonstrated mRNA delivery efficiency superior to that of lipid nanoparticles in primary human cells. Intramuscular injection of EVs carrying mRNA encoding SARS-CoV2 spike protein resulted in efficient mRNA expression near the injection site and resulted in sustained immune response to the viral protein. Additionally, no attenuation of mRNA expression was observed after re-administration of the exosomes [224]. Similar results were shown in another study at an in vivo model. EV-based siRNA delivery system was able to outperform lipid nano-particles for breast cancer siRNA delivery [225]. Apart from great biocompatibility and delivery efficiency, some types of EVs possess inherent targeting capabilities. For ex-ample, exosomes, derived from tumor cells are known to homotypically target tumor cells of the same type [226]. Additionally, the surface of EVs can be further modified with targeting and auxiliary molecules, which enhances the efficiency of EV-mediated delivery. In several studies, EVs decorated with "don't eat me" signals have demon-strated prolonged circulation in the bloodstream and reduced clearance by macrophages [227,228]. While decoration with targeting peptides results in targeted delivery of the cargo [228,229], development of cost-effective delivery system will require to combine several approaches during manufacturing. Cell lines that express EV secretion boosting molecules, packaging systems and targeting molecules will give the opportunity to achieve effective therapeutic effect with acceptable drug cost. As a summary, EVs have a potential to become “gold standard” of drug delivery vehicles in the future.”
Reviewer 3 Report
Comments and Suggestions for Authors
This is a very good review paper that covers the recent advances in extracellular vesicles. This review includes broad aspects and topics in the extracellular vesicle field. Further, the figures and schemes in this manuscript are very attractive. However, before accepting this manuscript, the authors should address some minor issues. The comments and suggestions are described as follows:
1. In Section 5, Challenges of loading cargo into EVs, it is suggested to provide more insights and perspectives into future research directions. The authors should add more details and discussion to support their claims. A good review paper should point out some key points for future study. Therefore, it is suggested that some forward-looking discussions be included on the emerging trends and future prospects of EV-based drug delivery systems.
2. It is suggested to enlarge the font sizes in the figures. Some figures are too blurry to be read.
3. In certain sections (e.g., methods for increasing EV yield or novel isolation techniques), the paper could cite more recent and emerging literature. This would ensure that readers have access to the most cutting-edge developments in the field.
4. It is suggested to expand the discussion comparing EVs to other drug delivery platforms, emphasizing clinical or preclinical outcomes that highlight EVs' unique advantages.
8. The authors are suggested to ensure consistent use of terms, such as EV subtypes and methods. For example, "exosomes" and "extracellular vesicles" are sometimes used interchangeably. Clear definitions in the early sections and consistent use throughout would reduce reader confusion.
Comments on the Quality of English Language
The authors should double-check the manuscript to eliminate typos and mistakes.
Author Response
Reviewer 3:
1) In Section 5, Challenges of loading cargo into EVs, it is suggested to provide more insights and perspectives into future research directions. The authors should add more details and discussion to support their claims. A good review paper should point out some key points for future study. Therefore, it is suggested that some forward-looking discussions be included on the emerging trends and future prospects of EV-based drug delivery systems.
Response 1: Thank you for this suggestion. Section 5 was supplemented with the following text: “Thus, each type of cargo requires special loading methods. In case of small mole-cules, application of endogenous methods results in low efficiency and possibly affects viability or metabolism of EV-producing cells. At the same time, the application of novel exogenous protocols based on ion gradients allows for high drug loading effi-ciency by utilizing active transport mechanisms, achieving up to 60% loading efficiency for doxorubicin [198]. Packaging of proteins with exogenous methods is complicated or can result in EV damaging (for example, during electroporation). Use of endogenous methods, such as viral-assisted loading or use of dimerization domains induce effec-tive packaging of desired proteins. For example, use of Nanoblades system for large Cas9 protein loading results in ~60 Cas9 proteins per vesicle [166]. In the context of RNA packaging, genetically engineered constructs enable efficient loading of RNA; however, they necessitate the introduction of substantial amounts of endogenous components. Chemical transfection of isolated EVs with cargo RNA, resulting in the formation of hybrid vesicles, presents a cost-effective alternative for small RNA mole-cules. Further experiments are needed to identify the most effective protocols for packaging various types of cargo.”
2) It is suggested to enlarge the font sizes in the figures. Some figures are too blurry to be read.
Response 2: We apologize for the oversight. Figures were corrected.
3) In certain sections (e.g., methods for increasing EV yield or novel isolation techniques), the paper could cite more recent and emerging literature. This would ensure that readers have access to the most cutting-edge developments in the field.
Response 3: Additional methods with references were added into section describing methods for increasing EV yield. Following text was introduced: “Alternative approaches to enhancing extracellular vesicle (EV) yield include electrical stimulation and biological transfection. These methods enable the scaling up of EV production while minimizing severe damage to producer cells by stimulating the ex-pression of genes that influence EV production. Cellular nanoelectroporation [75] and biological transfection [76] can also provide high endogenous loading of cargo such as functional mRNA.”; “Culturing of cells as a spheroids, use of hollow-fiber bioreactors, semi-permeable capsules, culturing on microcarriers in stirred tank bioreactors and rational media supplementation (increased glucose, additional nutrients and others) boosts EV yield up to 100-fold compared to standard 2-dimensional culturing.”
For isolation techniques, references to combined approaches were added: “As an optimal option, conditioned media after 3-dimentional culturing of cells in bioreactor can be processed by tandem isolation approaches such as UF-SEC or TFF-SEC to combine high and scalable yield of UF or TFF with effective elimination of main contaminants by SEC.”
“The methods described are well-established through years of EV research, and many of them demonstrate significant potential for scalability, including UF, TFF, and chromatography techniques. In recent years, novel methods for EV isolation for diagnostic purposes have been developed, including microfluidic approaches [67–69], assymetrical flow-field-flow fractionation [70], electrophoretic isolation [71], acoustic approaches [72] and others. Despite the high recovery rate, novel methods are not sutable for large-scale manufacturing and therefore are not considered in this review. Novel methods are descibed in detail in [35,73].”
- It is suggested to expand the discussion comparing EVs to other drug delivery platforms, emphasizing clinical or preclinical outcomes that highlight EVs' unique advantages.
Response 4: We appreciate your suggestion. The Discussion section was modified as follows: “Overall, extracellular vesicles represent a promising platform for drug delivery. In the context of cargo delivery, EVs offer nearly unmatched biocompatibility. They are char-acterized by prolonged circulation times, low toxicity, and reduced immunogenicity compared to other nanoplatforms, such as liposomes [223]. EVs have also demonstrated mRNA delivery efficiency superior to that of lipid nanoparticles in primary human cells. Intramuscular injection of EVs carrying mRNA encoding SARS-CoV2 spike protein resulted in efficient mRNA expression near the injection site and resulted in sustained immune response to the viral protein. Additionally, no attenuation of mRNA expression was observed after re-administration of the exosomes [224]. Similar results were shown in another study at an in vivo model. EV-based siRNA delivery system was able to outperform lipid nanoparticles for breast cancer siRNA delivery [225]. Apart from great biocompatibility and delivery efficiency, some types of EVs possess inherent targeting capabilities. For example, exosomes, derived from tumor cells are known to homo-typically target tumor cells of the same type [226]. Additionally, the surface of EVs can be further modified with targeting and auxiliary molecules, which enhances the effi-ciency of EV-mediated delivery. In several studies, EVs decorated with "don't eat me" signals have demonstrated prolonged circulation in the bloodstream and reduced clearance by macrophages [227,228]. While decoration with targeting peptides results in targeted delivery of the cargo [228,229], development of cost-effective delivery system will require to combine several approaches during manufacturing. Cell lines that ex-press EV secretion boosting molecules, packaging systems and targeting molecules will give the opportunity to achieve effective therapeutic effect with acceptable drug cost. As a summary, EVs have a potential to become “gold standard” of drug delivery vehicles in the future. ”
- The authors are suggested to ensure consistent use of terms, such as EV subtypes and methods. For example, "exosomes" and "extracellular vesicles" are sometimes used interchangeably. Clear definitions in the early sections and consistent use throughout would reduce reader confusion.
Response 8: Thank you, the terms were checked.
Round 2
Reviewer 1 Report
Comments and Suggestions for Authors
the paper can be published in the current form